

# Single-image super-resolution reconstruction based on phase-aware visual multi-layer perceptron (MLP)

Changteng Shi, Mengjun Li and Zhiyong An

Shandong Technology and Business University, Yantai, China

## ABSTRACT

Many advanced super-resolution reconstruction methods have been proposed recently, but they often require high computational and memory resources, making them incompatible with low-power devices in reality. To address this problem, we propose a simple yet efficient super-resolution reconstruction method using waveform representation and multi-layer perceptron (MLP) for image processing. Firstly, we partition the original image and its down-sampled version into multiple patches and introduce WaveBlock to process these patches. WaveBlock represents patches as waveform functions with amplitude and phase and extracts representative feature representations by dynamically adjusting phase terms between tokens and fixed weights. Next, we fuse the extracted features through a feature fusion block and finally reconstruct the image using sub-pixel convolution. Extensive experimental results demonstrate that SRWave-MLP performs excellently in both quantitative evaluation metrics and visual quality while having significantly fewer parameters than state-of-the-art efficient super-resolution methods.

## INTRODUCTION

Super-resolution reconstruction is a fundamental task in the field of image processing, aiming to recover high-resolution details from low-resolution ones to enhance image quality and detail presentation. In practical applications, super-resolution reconstruction holds significant importance in areas such as image enhancement, video compression, medical image processing and surveillance.

In recent years, there has been significant progress in super-resolution reconstruction technology, and researchers have been exploring new methods to enhance the quality of image reconstruction. Early methods based on convolutional neural networks (*Dong et al., 2014*; *Kim, Lee & Lee, 2016*; *Lim et al., 2017*; *Zhang et al., 2018*) used deep convolutional layers to perform super-resolution tasks, achieving certain results. However, they still have limitations in capturing details and overall image quality.

With the rise of Transformer architecture (*Liang et al., 2021*; *Liu et al., 2021b*), its powerful self-attention mechanism has provided new insights into super-resolution reconstruction. The efficiency and flexibility demonstrated by Transformers in handling sequential data allow them to capture richer contextual information in super-resolution

Corresponding author
Mengjun Li, lmj@sdtbu.edu.cn

tasks. However, the complex network structure of Transformers also brings significant computational and memory overhead, limiting their application in resource-constrained scenarios.

To further enhance the perceptual quality of super-resolution reconstruction, some researchers have proposed perceptually guided optimization objective estimation methods, such as SROOE (*Park, Moon & Cho, 2023*). These methods optimize perceptually relevant loss functions, enabling the generated super-resolution images to maintain clarity while also exhibiting better perceptual quality. However, there is still room for improvement in terms of computational efficiency and model complexity for these methods to adapt to a wider range of application scenarios.

In order to reduce computational burden, researchers have explored various methods, including efficient block design (*Michelini, Lu & Jiang, 2022*; *Kong et al., 2022*; *Sun, Pan & Tang, 2022*; *Li et al., 2022*; *Zhao et al., 2020*), knowledge distillation (*He et al., 2020*), neural architecture search (*Chu et al., 2021*), and structural re-parameterization (*Zhang, Zeng & Zhang, 2021*), to improve the efficiency of super-resolution algorithms. One important direction is to speed up inference time. Techniques such as sub-pixel convolution and model quantization have significantly accelerated runtime, while structural parameterization has improved the speed of the model during inference. However, these methods often sacrifice reconstruction performance for faster runtime. Therefore, there is still room for further exploration to find a better balance between model efficiency and reconstruction performance.

To overcome the above problems, we are inspired by the article on WaveMLP (*Tang et al., 2022*) and propose a new super-resolution reconstruction method SRWave-MLP. SRWave-MLP makes full use of the advantages of waveform representation and multi-layer perceptron, introduces an efficient WaveBlock in the processing process, and optimizes image feature extraction through the Token Mixing block and channel attention mechanism. At the same time, we design a downsampling residual mechanism to supplement some feature details for the model by processing the downsampled image of the original image. To this end, we also design a feature fusion block to achieve feature fusion of input data through a gating mechanism, so as to better fuse the original image features and the downsampled image features. Compared with previous methods, SRWave-MLP has fewer parameters, higher computational efficiency, and has achieved significant improvements in image quality.

Experimental results show that SRWave-MLP performs well in image super-resolution reconstruction tasks. Compared with other methods, SRWave-MLP is not only more efficient in computing resources, but also has achieved significant improvements in image quality. In addition, the SRWave-MLP model contains only 254K parameters and achieves relatively excellent super-resolution reconstruction results. We find that the SRWave-MLP model can achieve a better trade-off between SR performance and model complexity, as shown in Fig. 1.

The SRWave-MLP method has obtained superior performance in the field of super-resolution reconstruction. By taking full advantage of waveform representation and multi-layer perception to optimize the process of image feature extraction and reconstruction,

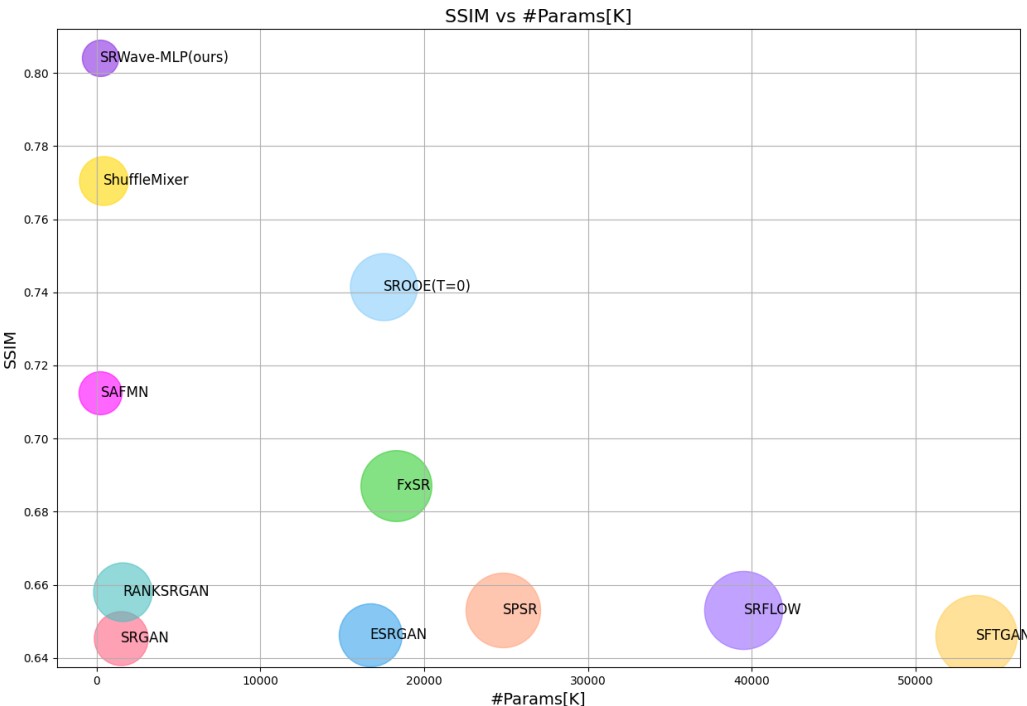

**Figure 1 Model complexity and structural similarity index measure (SSIM) comparison between our proposed SRWave-MLP model and other methods on BSDS100 for ×4 SR.** Circle sizes indicate the number of parameters. The proposed method achieves a better trade-off between model complexity and reconstruction performance.

SRWave-MLP brings a new solution for super-resolution reconstruction tasks. It provided a valuable reference for subsequent research in terms of image reconstruction quality and detail retention.

In summary, our contributions are as follows:

- Optimization of the WaveBlock was performed. Utilizing Token Mixing Blocks and channel attention mechanism enhanced the extraction of image features.
- The down-sampling residual is introduced to further improve the reconstruction quality by better recovering the lost detail information in the low-resolution image during the reconstruction process.
- A feature fusion block is designed to realize feature fusion and reconstruction of input data, thus improving the performance of image processing tasks.
- We quantitatively and qualitatively evaluate the proposed method on benchmark datasets and show that our SRWave-MLP achieves a good trade-off between accuracy and model complexity.

## RELATED WORKS

### CNN

In the super-resolution domain, convolutional neural network (CNN) is an important deep learning model that has made remarkable progress. Since the proposal of super-resolution convolutional neural network (SRCNN) (*Dong et al., 2014*), many CNN-based super-resolution methods have emerged. These methods mainly focus on three aspects: network structure design, loss function design and optimization strategy design.

In terms of network structure design, researchers have proposed many deep network architectures to improve the quality of super-resolution reconstruction. VDSR (*Kim, Lee & Lee, 2016*) uses a deeper network to enhance the representation ability of the model. EDSR (*Lim et al., 2017*) and RCAN (*Zhang et al., 2018*) introduce residual blocks and attention mechanisms to capture more important feature information.

In order to improve the perception quality of reconstructed images, the researchers introduced the perception loss function. ESRGAN (*Wang et al., 2018b*) uses a training method that combines perceptual loss and adversarial loss to produce more realistic reconstruction results.

In terms of optimization strategies, the researchers also explored different methods to improve training efficiency and model stability. For example, using gradient clipping and learning rate preheating in network training can accelerate convergence and improve generalization ability.

Overall, CNN's research in the field of super-resolution shows that deep learning brought breakthrough performance improvements to image super-resolution tasks. With the continuous progress of technology, we can foresee that more innovative CNN architectures and optimization strategies will be proposed in the future to promote the development of super-resolution reconstruction.

### Transformer

Transformer (*Vaswani et al., 2017*) is a model widely used in natural language processing and computer vision tasks, In recent years, many Transformer-based methods have been developed for advanced visual tasks such as image classification (*Liu et al., 2021b*; *Ramachandran et al., 2019*; *Dosovitskiy et al., 2020*; *Wu et al., 2020*; *Li et al., 2021b*; *Liu et al., 2021a*; *Vaswani et al., 2021*), object detection (*Liu et al., 2021b*; *Carion et al., 2020*; *Liu et al., 2020*; *Touvron et al., 2021*), and segmentation (*Liu et al., 2021b*; *Wu et al., 2020*; *Zheng et al., 2021*; *Cao et al., 2022*). Although visual Transformer has demonstrated the ability to model remote dependencies (*Dosovitskiy et al., 2020*; *Raghu et al., 2021*), studies have shown that combining convolution operations with Transformer can lead to better visual representations (*Li et al., 2023*; *Wu et al., 2021*; *Xiao et al., 2021*; *Yuan et al., 2021a*; *Yuan et al., 2021b*).

The Transformer architecture is also used for low-level visual tasks (*Chen et al., 2021*; *Li et al., 2021a*; *Liang et al., 2022*; *Liang et al., 2021*; *Tu et al., 2022*; *Wang et al., 2022*; *Zamir et al., 2022*). For example, IPT (*Chen et al., 2021*) introduced VIT-style networking and image processing through multi-task pre-training. Inspired by other studies, SwinIR (*Liang et al., 2021*), proposed an image recovery Transformer. VRT (*Liang et al., 2022*)

applied Transformer-based networking to video recovery tasks. EDT (*Li et al., 2021a*) used self-attention mechanism and multi-related task pre-training strategy to further promotes the latest technology of super-resolution reconstruction.

In this article, our approach differs from the previous Transformer-based approach in that it employed phase-aware vision multi-layer perceptron (MLP) as the basic network structure and utilized waveform representation and multi-layer perceptron network for image processing. This approach based on phase-aware vision can capture spatial information of images more comprehensively and learn complex patterns and relationships, resulting in better performance and results in super-resolution reconstruction tasks. By introducing phase-aware vision MLP in the field of super resolution, we aim to provide an entirely new solution for image reconstruction and achieve greater breakthroughs in performance and efficiency.

## Multi-layer perceptron

In recent years, with the rapid development of deep learning technology, the research and application of MLP has attracted more and more attention in the field of image super-resolution. The traditional super resolution method mainly uses interpolation algorithm or simple linear model, which is difficult to recover the complex high-frequency details in the image, resulting in the lack of realism and clarity in the reconstruction results. In contrast, MLP-based methods have powerful nonlinear mapping and learning capabilities, and are better able to capture rich texture and detail information in images, resulting in higher-quality super-resolution reconstructions.

Many researches have shown that MLP is no longer confined to simple encode-decode structure, but involve more complex network architecture in the field of super-resolution reconstruction. For example, ESPCN (*Shi et al., 2016*) used MLP layers for sub-pixel convolution. It mapped a low-resolution image to a higher-resolution space, the clarity and realism of the reconstruction are improved while preserving the image structure. This MLP architecture of WAN (*Yu et al., 2018*) enables efficient and accurate image super-resolution reconstruction through special activation functions, emphasizing the potential of MLPs in capturing image features. The CARN (*Ahn, Kang & Sohn, 2018*) model, which used a series of cascaded residual blocks (similar to MLP structures) to achieve fast image super-resolution reconstruction.

However, MLP still faces some challenges in the field of super resolution. Due to the large number of parameter requirements, the model is larger and the computing resource consumption is higher. At the same time, how to further improve the MLP network structure to increase performance while striking a balance between preserving detail and perceived quality is still worthy of in-depth study. In this study, we combined waveform representation and multi-layer perceptron networks to address the challenges of MLP in super-resolution reconstruction. Compared to previous MLP-based approaches, our approach has more advantages and innovation.

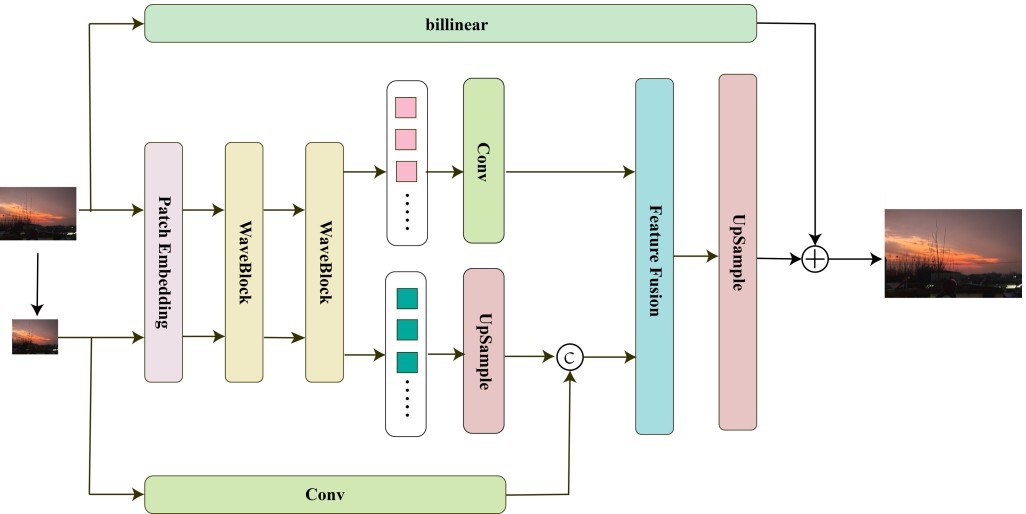

**Figure 2** SRWave-MLP structure.

# METHODOLOGY

## SRWave-MLP model structure

The SRWave-MLP model is shown in Fig. 2. The model consists of three parts: feature extraction, feature mixing, and image reconstruction.

Specifically, we first perform a bilinear downsampling operation on the low-resolution image $I_{LR} \in \mathbb{R}^{H \times W \times 3}$ to obtain the downsampled image $\widetilde{I}_{LR} \in \mathbb{R}^{\frac{H}{2} \times \frac{W}{2} \times 3}$. To make the model more compatible with computer vision tasks, we decided to use a feature map of shape $H \times W \times C$ to preserve the 2D spatial shape of the input image, where $H$, $W$, and $C$ represent the height, width, and number of channels, respectively. Therefore, we input $I_{LR}$ and $\widetilde{I}_{LR}$ into the Patch Embedding block respectively to obtain feature maps $M \in \mathbb{R}^{H \times W \times C}$ and $\widetilde{M} \in \mathbb{R}^{\frac{H}{2} \times \frac{W}{2} \times C}$. Then, WaveBlock is used to process them separately to extract features. The formula for the whole process is as follows:

$$M = Patch(I_{LR}), \widetilde{M} = Patch(\widetilde{I}_{LR}) \tag{1}$$

$$I'_{LR} = WaveBlock(M), \widetilde{I}'_{LR} = WaveBlock(\widetilde{M}). \tag{2}$$

Here $I'_{LR} \in \mathbb{R}^{H \times W \times C}$ and $\widetilde{I}'_{LR} \in \mathbb{R}^{\frac{H}{2} \times \frac{W}{2} \times C}$ represent the processed $M$ and $\widetilde{M}$ respectively.

Afterward, we process $I'_{LR}$ and $\widetilde{I}'_{LR}$ separately. For $I'_{LR}$, we further process it using a $3 \times 3$ convolution to extract more representative features. For $\widetilde{I}'_{LR}$, we first perform upsampling using an upsampling block. Then, we double its channel number to $2C$ using a $1 \times 1$ convolution. Next, we perform upsampling using sub-pixel convolution. Finally, we concatenate it with the features simply extracted by a $3 \times 3$ convolution from $I_{LR}$ to

further supplement the lost details. The formula for the whole process is as follows:

$$I''_{LR} = Conv_{3 \times 3}(I'_{LR}) \tag{3}$$

$$\widetilde{I}''_{LR} = Cat(Pixel(Conv_{1 \times 1}(\widetilde{I}'_{LR})), Conv_{3 \times 3}(I_{LR})) \tag{4}$$

Here $I''_{LR} \in \mathbb{R}^{H \times W \times C}$, $\widetilde{I}''_{LR} \in \mathbb{R}^{H \times W \times C}$.

The feature $\widetilde{I}''_{LR}$ we extracted from the downsampled image can effectively make up for the missing details in $I''_{LR}$. We send them to the feature fusion block for feature fusion to obtain $F_{LR} \in \mathbb{R}^{H \times W \times C}$. The feature fusion block can adaptively select and fuse the key information in the input data, thereby achieving more accurate feature extraction and better image reconstruction. Finally, the upsampling block composed of convolution and PixelShuffle layers is used to reconstruct the image, and the residual connection is made with the image upsampled by billinear to obtain the final reconstructed image $I_{SR}$. The formula for the whole process is as follows:

$$F_{LR} = Fusion(I''_{LR}, \widetilde{I}''_{LR}) \tag{5}$$

$$I_{SR} = billinear(I_{LR}) + Pixel(Conv_{3 \times 3}(F_{LR})) \tag{6}$$

---

**Algorithm 1** Super-Resolution Reconstruction using SRWave-MLP

---

1: function SRWave-MLP(x);
2: **Input:** Low-resolution image $I_{LR}$, model parameters
3: **Output:** Super-resolution image $I_{SR}$
4: $\widetilde{I}_{LR} \leftarrow Bilinear(I_{LR})$
5: $M, \widetilde{M} \leftarrow PatchEmbedding(I_{LR}, \widetilde{I}_{LR})$
6: **for** $i \leftarrow 1$ to $n = 2$ **do**
7: $\quad I'_{LR}, \widetilde{I}'_{LR} \leftarrow WaveBlock(M, \widetilde{M})$
8: **end for**
9: $I''_{LR} \leftarrow Conv_{3 \times 3}(I'_{LR})$
10: $\widetilde{I}''_{LR} \leftarrow Concatenate(Pixle(Conv_{1 \times 1}(\widetilde{I}'_{LR})), Conv_{3 \times 3}(I_{LR}))$
11: $F_{LR} \leftarrow Fusion(I''_{LR}, \widetilde{I}''_{LR})$
12: $I_{SR} \leftarrow Billinear(I_{LR}) + Pixel(Conv_{3 \times 3}(F_{LR}))$
13: **return** $I_{SR}$

---

## WaveBlock

WaveBlock is a block we introduced from WaveMLP (*Tang et al., 2022*). We have made some improvements to it to make it suitable for super-resolution reconstruction of images.

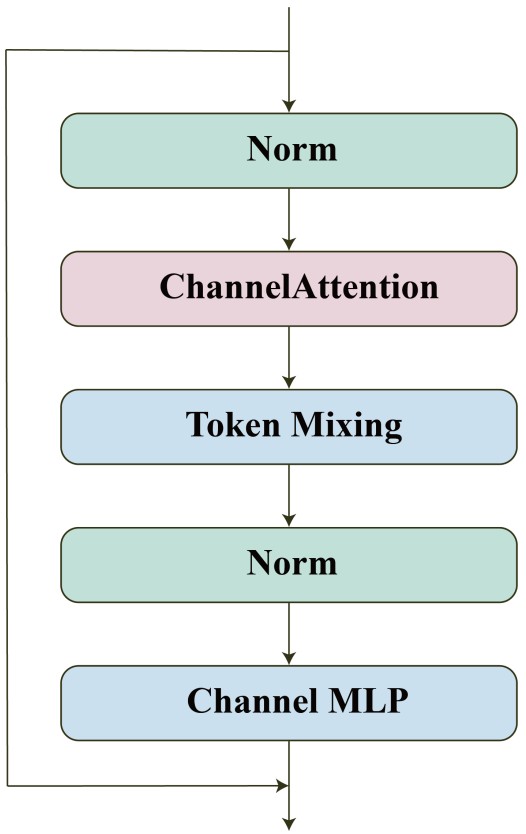

**Figure 3  WaveBlock.**

As shown in Fig. 3, we introduced the channel attention mechanism and removed the downsampling block in the original model. The channel attention mechanism allows the model to automatically learn the importance between different channels, thereby improving the ability of feature representation. Removing the downsampling block avoids excessively reducing the resolution of the feature map, thereby avoiding the loss of more detailed information. Such modifications may help improve the efficiency and performance of the model.

In addition to channel attention, WaveBlock also contains two other important parts: Channel MLP and Token Mixing blocks. Channel MLP is composed of a simple stack of FC layers and nonlinear activation functions. Token Mixing consists of two phase-aware token mixing modules (PATMs), which use amplitude and phase information to aggregate information between different tokens.

PATM uses a phase-aware mechanism to enhance the representation ability of features. First, through phase-aware convolution, the model can extract position information from the input features, so that the model has a better understanding of the spatial structure of the input. The formula is as follows:

$$\theta_h = Conv_{phase}(x), \theta_w = Conv_{phase}(x) \tag{7}$$

Next, FC is used to extract the feature representations of height, width, and channel respectively. The formula is as follows:

$$x_h = FC_h(x), x_w = FC_w(x), x_c = FC_c(x) \tag{8}$$

Then, the cosine and sine functions are used to modulate the features and fuse the features with the position information to enhance the model's sensitivity to position. The formula is as follows:

$$x'_h = Cat(x_h \cdot cos\theta_h, x_h \cdot sin\theta_h) \tag{9}$$

$$x'_w = Cat(x_w \cdot cos\theta_w, x_w \cdot sin\theta_w) \tag{10}$$

Next, the modulated features are processed using the FC layer to obtain phase-aware features in height and width. The formula is as follows:

$$h = FC_h(x'_h), w = FC_w(x'_w) \tag{11}$$

Subsequently, the features in height, width, and channel are adaptively fused through adaptive average pooling and multi-layer perceptrons, and the attention weights are calculated. The formula is as follows:

$$a = AvgPool(h + w + x_c) \tag{12}$$

$$\alpha = reshape_{B,C,3}(MLP(a)) \tag{13}$$

Finally, the model performs weighted fusion of the features according to the attention weights to obtain the final feature representation. The formula is as follows:

$$x' = \alpha[0] \cdot h + \alpha[1] \cdot w + \alpha[2] \cdot c \tag{14}$$

---

**Algorithm 2** PATM

1: function PATM(x);
2: **Input:** feature tensor $x$
3: **Output:** feature tensor $x$
4: $\theta_h \leftarrow Conv_{1\times1}(x)$
5: $\theta_w \leftarrow Conv_{1\times1}(x)$
6: $x_h \leftarrow FC_h(x), x_w \leftarrow FC_w(x)$
7: $x'_h \leftarrow Concatenate(x_h * cos(\theta_h), x_h * sin(\theta_h))$
8: $x'_w \leftarrow Concatenate(x_w * cos(\theta_w), x_w * sin(\theta_w))$
9: $h \leftarrow FC(x'_h), w \leftarrow FC(x'_w), c \leftarrow FC(x)$
10: $a \leftarrow AvgPool2d(h + w + c)$
11: $a \leftarrow MLP(a)$
12: $a \leftarrow reshape_{B,C,3}(a)$
13: $x \leftarrow h * a[0] + w * a[1] + c * a[2]$
14: **return** $x$

---

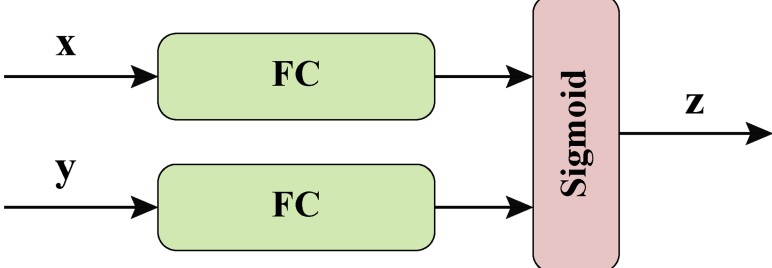

**Figure 4** **Feature fusion block.**

## Feature fusion block

The feature fusion block is a key component of our neural network model. The key components of the neural network model are shown in Fig. 4.

Firstly, an intermediate representation $Z_1$ is obtained through the first fully connected layer, the equation is as follows:

$$Z_1 = FC(x) \tag{15}$$

Then another intermediate representation $Z_2$ is obtained through the second fully connected layer, the equation is as follows:

$$Z_2 = FC(y) \tag{16}$$

Next, the sum of $Z_1$ and $Z_2$ is input into the Sigmoid function to obtain the fused intermediate representation $Z$, the equation is as follows.

$$Z = Sigmoid(Z_1 + Z_2) \tag{17}$$

Finally, the input data sum is reconstructed according to the fused intermediate representation through a linear combination operation. Specifically, by calculating the element-level product of $1 - Z$ and $Z$, we get the contribution to $x$. Meanwhile, by calculating the element-level product of $y$ and $Z$, we get the contribution to $y$. Finally, the contributions of the two parts are added together to get the output of the feature fusion block *res*. The equation is as follows:

$$res = (1 - Z)x + Zy. \tag{18}$$

The design idea of feature fusion block is to realize the fusion and reconstruction of input data through the gating mechanism, so as to extract more abundant feature information. Through the combination of multiple fully connected layers and activation function Sigmoid, the feature fusion block can learn complex patterns and relationships of input data to improve the performance of image processing tasks. Through the introduction of gating mechanism, the feature fusion block can adaptively select and fuse the key information in the input data, so as to achieve more accurate feature extraction and reconstruction.

The parameters and hyperparameters of the block can be adjusted and optimized for specific tasks. In the experiment, we will verify the performance of the feature fusion block in image processing tasks and compare it with other methods to verify its effectiveness and superiority.

---

**Algorithm 3** Feature Fusion

---

1: function Fusion(x,y);
2: **Input:** feature tensor $x$, feature tensor $y$
3: **Output:** feature tensor $res$
4: $Z_1 \leftarrow FC(x)$
5: $Z_2 \leftarrow FC(y)$
6: $Z \leftarrow Sigmoid(Z_1 + Z_2)$
7: $res \leftarrow (1 - Z)x + Zy$
8: **return** $res$

---

# EXPERIMENT AND ANALYSIS

## Experiment settings

We conduct experiments using DF2K (DIV2K (*Agustsson & Timofte, 2017*) + Flicker2K (*Timofte et al., 2017*)) and DIV2K (*Agustsson & Timofte, 2017*) datasets, which contain 2,650 and 800 high-quality natural images, respectively. Our test data sets include BSDS100 (*Martin et al., 2001*), General100 (*Dong, Loy & Tang, 2016*), Urban100 (*Huang, Singh & Ahuja, 2015*), Manga109 (*Matsui et al., 2017*) and DIV2K (*Agustsson & Timofte, 2017*) verification sets, and the image contents are mainly people, animals and natural landscape images in different scenes. In order to prevent overfitting to some extent, the training image is enhanced in the process of image preprocessing. Data enhancement can extract more information from the original data set, thereby narrowing the gap between the training set and the validation set. The specific operation is to rotate these images randomly 90°, 180°, 270° and then flip them horizontally to obtain an enhanced dataset. We implemented our super-resolution reconstruction model based on the PyTorch deep learning framework. We used a method based on deep convolutional neural networks for reconstruction. We used the Adam optimizer (*Kingma & Ba, 2014*) to train the model, and the initial learning rate was set to 0.01. We used a batch size of 16 and trained 400 epochs during the training process.

---

**Algorithm 4** Training

1: function train();
2: **Input:** Training data $(X, Y)$, learning rate $\eta = 0.01$, number of epochs $N = 400$, batch size $B = 16$
3: **Output:** Trained model parameters $\theta$
4: Initialize model parameters $\theta$ randomly
5: **for** $epoch \leftarrow 1$ to $N$ **do**
6:     **for** $(x_i, y_i) \in (X, Y)$ **do**
7:         Compute model prediction $\hat{y}_i = \text{Model}(x_i, \theta)$
8:         Compute loss $L = \text{Loss}(y_i, \hat{y}_i)$
9:         Compute gradients $\nabla_\theta L$
10:         Update parameters $\theta \leftarrow \theta - \eta \cdot \nabla_\theta L$
11:     **end for**
12: **end for**
13: **return** Trained model parameters $\theta$

---

## Result analysis

Peak signal-to-noise (PSNR) and structural similarity index measure (SSIM) were performed by SRWave-MLP reconstruction algorithm with 10 reconstruction algorithms. The SRWave-MLP reconstruction algorithm is compared with 10 reconstruction algorithms including SRGAN (*Ledig et al., 2017*), ESRGAN (*Wang et al., 2018b*), SFTGAN (*Wang et al., 2018a*), RankSRGAN (*Zhang et al., 2019*), SRFlow (*Lugmayr et al., 2020*), SPSR (*Ma et al., 2020*), FxSR (*Park, Moon & Cho, 2022*), ShuffleMixer (*Sun, Pan & Tang, 2022*), SROOE (*Park, Moon & Cho, 2023*) and SAFMN (*Sun et al., 2023*) on the DIV2K (*Agustsson & Timofte, 2017*), BSDS100 (*Martin et al., 2001*), General100 (*Dong, Loy & Tang, 2016*), Manga109 (*Matsui et al., 2017*), and Urban100 (*Huang, Singh & Ahuja, 2015*) datasets in terms of PSNR and SSIM. The comparison results are shown in Table 1. With fewer parameters than other super-resolution methods, the SRWave-MLP reconstruction algorithm achieves the best results in terms of SSIM and performs well in terms of PSNR.

In addition to the quantitative metrics, we also compared the visual results of the reconstructed images of different methods. Figure 5 shows the visual comparison of low-resolution images of some example images and the reconstructed images of different methods. We can observe that SRWave-MLP generates more accurate structures and details.

We compare the parameter sizes of existing super-resolution reconstruction algorithms and our proposed SRWave-MLP, and conduct a comprehensive analysis of the performance of these algorithms based on the PSNR and SSIM results on the BSDS100 test set in the previous experiments. From Table 1 and Fig. 6, we can observe that there are significant differences in the parameter sizes of different algorithms. In our SRWave-MLP model, the number of parameters is only 235K, which is much lower than SROOE's 17500K, FxSR's 18300K, SPSR's 24800K and SRFLOW's 39500K. Our model is able to achieve

**Table 1 Comparison with state-of-the-art SR methods on benchmarks.** The first and second best results in each group are highlighted in bold.

| Model | Venue | Params (K) | DIV2K | | BSDS100 | | General100 | | Manga109 | | Urban100 | |
|---|---|---|---|---|---|---|---|---|---|---|---|---|
| | | | PSNR | SSIM | PSNR | SSIM | PSNR | SSIM | PSNR | SSIM | PSNR | SSIM |
| SRGAN | CVPR2017 | 1,500 K | 26.63 | 0.7625 | 24.13 | 0.6454 | 27.54 | 0.7998 | 26.26 | 0.8285 | 22.84 | 0.7196 |
| ESRGAN | ECCV2018 | 1,6700 K | 26.64 | 0.7640 | 23.95 | 0.6463 | 27.53 | 0.7984 | 26.50 | 0.8245 | 22.78 | 0.7214 |
| SFTGAN | CVPR2018 | 53,700 K | 26.56 | 0.7578 | 24.09 | 0.6460 | 27.04 | 0.7861 | 26.07 | 0.8182 | 22.74 | 0.7107 |
| RANKSRGAN | ICCV2019 | 1,600 K | 26.51 | 0.7526 | 24.09 | 0.6580 | 27.31 | 0.7899 | 26.04 | 0.8117 | 22.93 | 0.7169 |
| SRFLOW | ECCV2020 | 39,500 K | 27.08 | 0.7558 | 24.66 | 0.6531 | 27.83 | 0.7951 | 27.11 | 0.8244 | 23.68 | 0.7316 |
| SPSR | CVPR2020 | 24,800 K | 26.71 | 0.7614 | 24.16 | 0.6531 | 27.65 | 0.7995 | 26.74 | 0.8267 | 23.24 | 0.7365 |
| FxSR | IEEE Access2022 | 18,300 K | 27.51 | 0.7890 | 24.77 | 0.6871 | 28.44 | 0.8229 | 27.64 | 0.8440 | 24.08 | 0.7641 |
| ShuffleMixer | NeurIPS2022 | 411 K | **29.57** | **0.8507** | **27.21** | **0.7706** | **30.04** | **0.8734** | **29.53** | **0.9036** | **25.10** | 0.7914 |
| SROOE ($T=0$) | CVPR2023 | 17,500 K | **29.33** | 0.8413 | 26.45 | 0.7416 | **30.08** | 0.8662 | **29.36** | 0.8948 | **25.21** | **0.8020** |
| SAFMN | ICCV2023 | 240 K | 28.97 | 0.8182 | 26.38 | 0.7125 | 29.97 | 0.8707 | 28.52 | 0.8772 | 24.79 | 0.7811 |
| SRWave-MLP (ours) | | 235 K | 29.02 | **0.8697** | 26.84 | **0.8041** | 29.34 | **0.8960** | 28.03 | **0.9098** | 24.22 | **0.8084** |

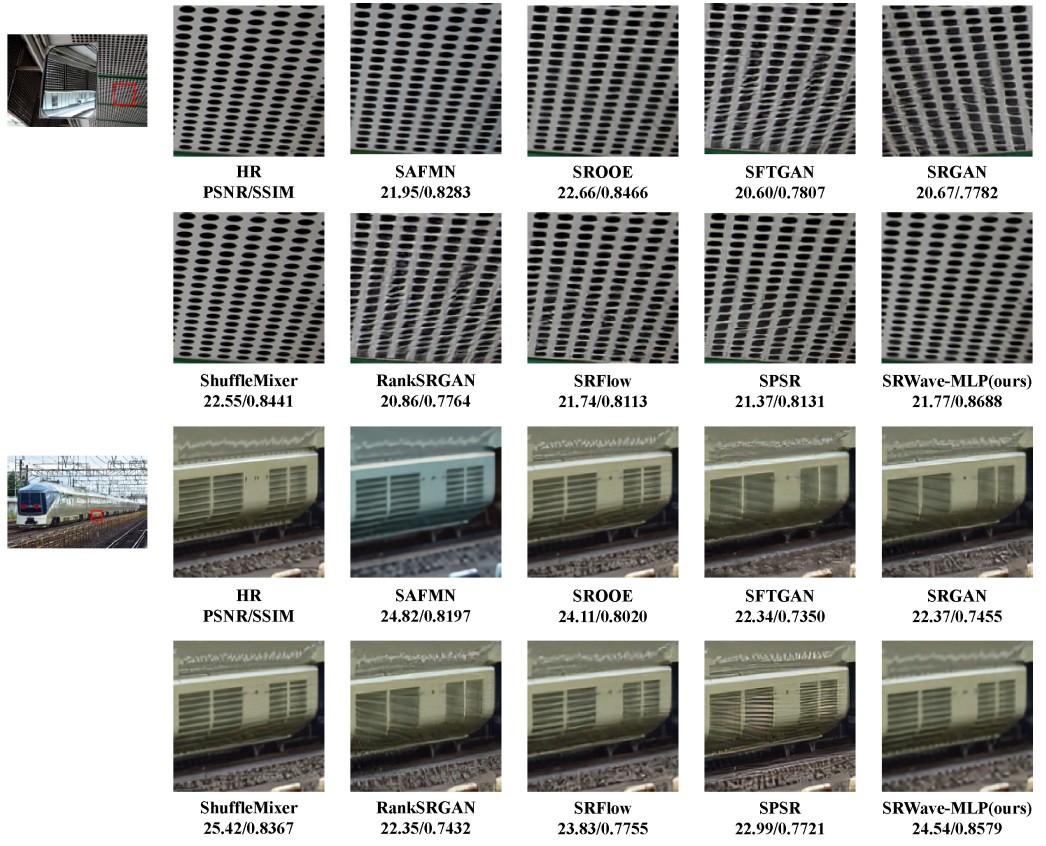

**Figure 5 Visual comparison with state-of-the-art SR methods on the DIV2K dataset.**

good performance while maintaining a small model size. In addition, when the number of parameters of our model is smaller than that of other lightweight models SAFMN and ShuffleMixer, the results on the SSIM are optimal and the PSNR performs well. In

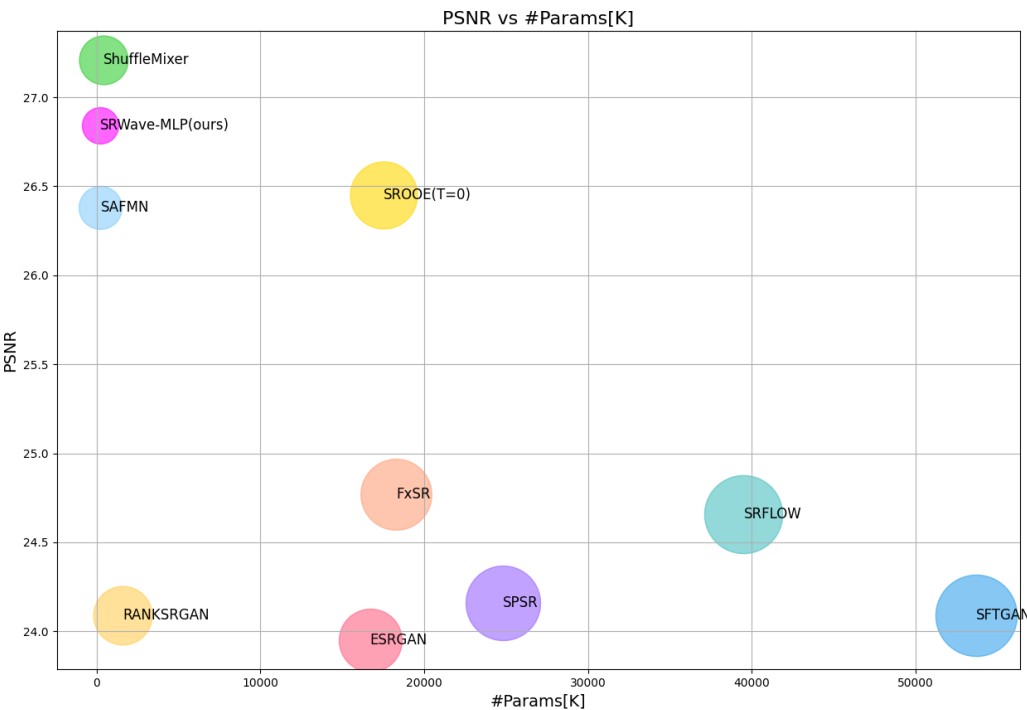

**Figure 6 Model complexity and PSNR comparison between our proposed SRWave-MLP model and other methods on BSDS100 for ×4 SR.** Circle sizes indicate the number of parameters. The proposed method achieves a better trade-off between model complexity and reconstruction performance.

super-resolution reconstruction tasks, there is a trade-off between parameter size and performance. Larger model parameters may mean more powerful learning capabilities and better image reconstruction quality, but also bring higher storage requirements and computational complexity. SRWave-MLP has obvious advantages over large models such as SROOE in terms of parameter size, and still achieving relatively excellent results in terms of performance. This makes SRWave-MLP a promising choice both in terms of image reconstruction quality and model lightweight.

In addition, the comparison results of our super-resolution reconstruction model SRWave-MLP with two Tranformer-based comparison models SWIR (*Liang et al., 2021*) and HAT (*Chen et al., 2023*) in terms of the number of parameters are shown in Table 2, combined with the PSNR and SSIM results of the BSDS100 (*Martin et al., 2001*) test set in the previous experiment to measure the performance of different models in terms of reconstructed image quality and structural similarity. Our model SRWave-MLP has the least number of parameters, only 0.24M, which is much less than the number of parameters of SWIR and HAT. This shows that our model has an advantage in model lightweight. Although our model has fewer parameters, it performs better than SWIR and HAT in terms of SSIM, reaching 0.8048. In terms of PSNR, although SRWave-MLP is slightly inferior to SWIR and HAT, it is still close to their results.

Our SRWave-MLP model has an advantage in terms of the number of parameters, while still achieving relatively good results in performance. Our results show that the lightweight

**Table 2  Comparison with SWIR and HAT in terms of parameter count.**

| Model | Parameter count (M) | PSNR | SSIM |
|---|---|---|---|
| SWIR | 12 | 27.92 | 0.7489 |
| HAT | 20 | 27.97 | 0.7505 |
| SRWave-MLP (ours) | 0.24 | 26.84 | 0.8041 |

**Table 3  Impact of feature fusion block.**

| Method | Fusion | Simple residual | Concatenation |
|---|---|---|---|
| PSNR | 26.84 | 26.65 | 26.54 |
| SSIM | 0.8041 | 0.7990 | 0.7983 |

**Table 4  Impact of subsampling residuals.**

| Method | With subsampling residual | Without subsampling residual |
|---|---|---|
| PSNR | 26.84 | 25.91 |
| SSIM | 0.8041 | 0.7416 |

SRWave-MLP model has great application potential in super-resolution reconstruction tasks, and can achieve excellent image reconstruction results while maintaining small model parameters.

## Ablation experiment

For the ablation study, we train SRWave-MLP on DF2K (*Lim et al., 2017*) of classical image SR($\times 4$) and test it on BSDS100 (*Martin et al., 2001*).

**The impact of the feature fusion block:** The influence of using mixed block, simple residual joining, and concatenation of different levels of feature tensors in channel dimensions on PSNR and SSIM are shown in Table 3. From the table, we can draw the following observations.

Firstly, the Fusion block is important because it improves by 0.19dB compared to using a simple residual connection PSNR, which indicates that the feature fusion block helps to improve the peak signal-to-noise ratio of the reconstructed image to a certain extent, that is, it increases the clarity and detail retention of the reconstructed image. Secondly, compared with the splicing of feature tensors of different levels in the channel dimension, the PSNR improves by 0.30 dB, which further demonstrates the effectiveness of Fusion block in improving the quality of reconstructed images. Compared with simple feature stitching, Fusion block can better fuse different levels of feature representation, avoid the loss of feature information, and thus improve the quality of the reconstructed image.

**The impact of downsampled residual operation:** The effects on PSNR and SSIM with and without down-sampled residuals are shown in Table 4. From the table, we can draw the following observations.

We can see that the PSNR improved by 0.93db with down-sampled residuals compared to PSNR without down-sampled residuals. The use of down-sampling residuals can bring significant PSNR improvement in super-resolution reconstruction tasks, while maintaining

**Table 5  Impact of channel attention.**

| Method | Without channel attention | With channel attention |
|--------|---------------------------|------------------------|
| PSNR | 26.84 | 26.70 |
| SSIM | 0.8041 | 0.8023 |

good image structure similarity. This confirms the validity of the down-sampling residual operation in super-resolution reconstruction tasks and its important role in improving model performance.

**The impact of channel attention in WaveBlock:** In this article, we introduced the channel attention mechanism into the WaveBlock to explore its impact on image processing. Through experimental comparison, we found that our method performs better in image reconstruction quality when using channel attention mechanism. As shown in Table 5, when the channel attention mechanism is used, the PSNR value of our method reaches 26.84 and the SSIM value reaches 0.8041. When the channel attention mechanism is not used, the PSNR value is slightly lower, 26.70, and the SSIM value is 0.8023.

The experimental results show that the introduction of channel attention mechanism can significantly improve the performance of our method in image processing tasks. The channel attention mechanism can adaptively adjust the weight of different channels, so that more critical and useful channel information can be more emphasized under a specific task. Therefore, the channel attention mechanism plays an important role in our approach, allowing us to capture the detailed features of the image more precisely, resulting in higher quality image reconstruction results.

**The impact of the number of MLP hidden layers in WaveBlock:** According to the experimental results in Table 6, we analyzed the influence of MLP hidden layers in WaveBlock on image processing tasks. When the number of MLP hiding layers is 2, the average PSNR of the network in the image reconstruction task is 26.69. This shows that the reconstruction performance of the network is relatively mediocre in the case of fewer layers. when we increase the MLP hidden layers to four layers, average PSNR increased to 26.84, shows a certain performance improvements. Increasing the number of hidden layers enables the network to extract image features at a deeper level, thus improving the PSNR index. However, as to further increase the number of six layer, the average PSNR decrease slightly to 26.76. This may indicate that in some cases, too many layers may introduce some unnecessary complexity, which can affect rebuild performance. In 8th layer, the average PSNR further reduce to 26.69, close to the initial layer 2 network performance. This indicates that increasing the number of hidden layers within a certain range can improve the performance of the network, but too many layers may cause performance degradation.In addition to PSNR, we also looked at SSIM metrics. When the number of MLP hidden layers is four, the network performs best and its SSIM value is 0.8041.

In summary, appropriately increasing the number of MLP hidden layers can improve the performance of the network, but too many layers may cause performance degradation. In practical applications, we need to choose the right number of MLP hiding layers according

**Table 6  Impact of number of hidden layers.**

| Number of layers | 2 | 4 | 6 | 8 |
|---|---|---|---|---|
| PSNR | 26.69 | 26.84 | 26.76 | 26.69 |
| SSIM | 0.8006 | 0.8041 | 0.8027 | 0.7992 |

**Table 7  Impact of number of layers in PATM block.**

| Number of layers | 2 | 3 | 4 |
|---|---|---|---|
| PSNR | 26.72 | 26.84 | 26.79 |
| SSIM | 0.8021 | 0.8041 | 0.8035 |

to the specific situation, in order to balance performance and complexity, and obtain the best image reconstruction quality.

**The impact of PATM block layers in WaveBlock:** According to the experimental results in Table 7, we analyzed the influence of PATM block layers on image processing tasks. When the number of PATM block is 2 layers, the average PSNR of the network in the image reconstruction task is 26.72. This shows that in the case of fewer layers, the network performs better in terms of reconstruction performance. When we increased the number of layers of the PATM block to 3, the average PSNR increased to 26.84. This suggests that increasing the number of layers of PATM block helps improve network performance and leads to higher rebuild quality. Compared with layer 2, Layer 3 PATM block enables the network to restore image details more accurately, thus improving the PSNR index. When the number of layers of PATM block was further increased to 4 layers, the average PSNR decreased slightly to 26.79. This may indicate that in some cases, too many layers may introduce some unnecessary complexity, which can affect rebuild performance.

In addition to the PSNR, we also examined the SSIM index. When the number of layers of PATM block is 3, the network performance is the best, and its SSIM value is 0.8041, slightly better than that of 2 layers (SSIM =0.8021) and 4 layers (SSIM =0.8035). This further verifies the effectiveness of the 3-layer PATM block in image processing tasks. To sum up, the appropriate increase PATM block layer helps to improve the reconstruction of the network performance, but too many layers may not always lead to better results. In practical applications, we need to choose the right number of PATM block layers according to the specific situation, in order to balance performance and complexity, and obtain the best image reconstruction quality.

## CONCLUSION

This article proposes a lightweight super-resolution reconstruction method aimed at extracting richer detailed features. Through the accurate extraction and reconstruction of image features through WaveBlock and feature fusion blocks, we are not only more efficient in computational resources, but also achieve significant improvements in image quality. Our method is a lightweight perception-oriented model. Compared with other large-scale models, our method is slightly insufficient in the PSNR evaluation metric, but

outperforms all others in the structural similarity (SSIM) evaluation metric. At the same time, our method has a small size and number of parameters, making it highly adaptable and deployable on devices with limited resources.

Although our model shows good performance in a lightweight model, it may have some limitations in handling long-distance dependencies compared to large Transformer-based models, which may cause the PSNR of the reconstructed image to be slightly lower than expected. To solve this problem, we can consider introducing adversarial training technology to further optimize the model by introducing adversarial loss functions to improve the model's ability to restore image details.

## ACKNOWLEDGEMENTS

I would like to thank the anonymous reviewers whose comments and suggestions helped improve this manuscript.

### Funding

This work was supported by the Natural Science Foundation Project of Shandong Province, China (No. ZR2021MF068). The funders had no role in study design, data collection and analysis, decision to publish, or preparation of the manuscript.

### Grant Disclosures

The following grant information was disclosed by the authors:
Natural Science Foundation Project of Shandong Province, China: No. ZR2021MF068.

### Competing Interests

The authors declare there are no competing interests.

### Author Contributions

- Changteng Shi conceived and designed the experiments, performed the computation work, prepared figures and/or tables, and approved the final draft.
- Mengjun Li performed the experiments, authored or reviewed drafts of the article, and approved the final draft.
- Zhiyong An analyzed the data, prepared figures and/or tables, and approved the final draft.

### Data Availability

The BSDS100: Berkeley Segmentation Data Set 100 is available at: shi, changteng . (2024). BSDS100 [Data set]. Zenodo. https://doi.org/10.5281/zenodo.11177719

shi, changteng . (2024). Single-image super-resolution reconstruction based on phase-aware visual MLP. Zenodo. https://doi.org/10.5281/zenodo.11177671

The DIV2K: DIVerse 2K resolution high-quality images dataset is available at: https://data.vision.ee.ethz.ch/cvl/DIV2K.

The General-100 dataset for image super-resolution is available at: http://mmlab.ie.cuhk.edu.hk/projects/FSRCNN.html.

The Urban-100 dataset for image super-resolution is available at: https://github.com/jbhuang0604/SelfExSR/tree/master/data.

The Manga109 dataset for image super-resolution is available at: http://www.manga109.org/en.

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
