# Peer review of "Single-image super-resolution reconstruction based on phase-aware visual multi-layer perceptron (MLP)"

_PeerJ Computer Science, doi:10.7717/peerj-cs.2208_

## Round 0.1 · original submission · Major Revisions

Please revise the paper according to the comments. Then it will be evaluated again.

Reviewer 1 ·

Basic reporting

1. The abstract and conclusion need to be improved. The abstract must be a concise yet comprehensive reflection of what is in your paper. Please modify the abstract according to “motivation, description, results and conclusion” parts. I suggest extending the conclusions section to focus on the results you get, the method you propose, and their significance.
2. What is the motivation of the proposed method? The details of motivation and innovations are important for potential readers and journals. Please add this detailed description in the last paragraph in section I. Please modify the paragraph according to "For this paper, the main contributions are as follows: (1) ......" to Section I. Please give the details of motivations. In Section 1, I suggest the authors can amend your contributions of manuscript in the last of Section 1.
3. The introduction section of the paper needs to revise according to the timeline of technology development. Please update references with recent papers in CVPR, ICCV, ECCV et al

Experimental design

4. Please give the details of proposed method for proposed model. I suggest the authors amend the calculation of your size of proposed method and the details is important for proposed method. In section 2, some equations need to improve related depict, for example, equation (1) and (2).
5. The manuscript needs to improve description for proposed method and innovations. The writing dos not meet the requirements for publication and I suggest the authors can improve the depict for effective paragraphs in your revised manuscript.
6. The content of experiments needs to amend related experiments to compare related SOTA in recent three years. I recommend the authors amend related experimental results of proposed method of SOTA according to the published paper in IEEE, Springer and Elsevier.

Validity of the findings

7. However, the manuscript, in its present form, contains several weaknesses. Adequate revisions to the following points should be undertaken in order to justify recommendation for publication.
8. In the conclusion section, the limitations of this study and suggested improvements of this work should be highlighted.
9. Please check all parameters in the manuscript and amend some related description of primary parameters. In section 3, please write the proposed algorithm in a proper algorithm/pseudocode format with section 3. Otherwise, it is very hard to follow. Some examples here: https://tex.stackexchange.com/questions/204592/how-to-format-a-pseudocode-algorithm

Additional comments

none

Reviewer 2 ·

Basic reporting

All comments have been added in detail to the 4th section called additional comments.

Experimental design

All comments have been added in detail to the 4th section called additional comments.

Validity of the findings

All comments have been added in detail to the 4th section called additional comments.

Additional comments

Review Report for PeerJ Computer Science
(Single-image super-resolution reconstruction based on phase- aware visual MLP)

1. Within the scope of the study, a super-resolution reconstruction approach with phase-aware visual Multi-Layer Perceptron architecture, called SRWave-MLP and consisting mainly of waveforms, features fusion and image reconstruction steps, has been proposed.

2. In the Introduction section, Super-resolution reconstruction and the importance of the subject are mentioned, and the main contributions of the study are clearly expressed.

3. In the Related works section, the literature related to super-resolution was examined in terms of convolutional neural networks, transformers and multi-layer perceptron networks. A literature table consisting of columns such as the model used, dataset, results, advantages and disadvantages can be added at the end of this section to more clearly emphasize the importance of the study and its differences from the literature.

4. In the methodology section, the SRWave-MLP model structure, WaveBlock and Feature Fusion Module recommended within the scope of the study are clearly mentioned.

5. The proposed model was used in the Berkeley Segmentation Data Set 100, General-100 dataset for image super-resolution, Manga109 dataset for image super-resolution, Urban-100 dataset for image super-resolution and DIVerse 2K resolution high-quality images datasets, which are frequently used in the literature. In addition, both generative adversarial network based and other state-of-the-art models were used to compare the results with other models. Testing the proposed model on many different open source datasets and comparing it with different current models is both positive and sufficient.

6. It should be explained how the parameters such as the optimizer used, batch size, learning rate were determined and whether different experiments were made.

7. It has been observed that models and codes are shared via the github platform. This will further increase both the contribution of the study to the literature and its post-publication usability.

8. The comparisons made and the structural similarity index measure (SSIM) and peak signal-to-noise ratio (PSNR) results of the proposed model are at an acceptable level.

As a result, although the study has the potential to contribute to the literature in terms of the proposed model and its originality, it is recommended to pay attention to the above-mentioned items.

---

## Round 0.2 · accepted · Accept

Thanks to the authors for their efforts to improve the work. The current version satisfied the reviewers successfully. It can be accepted now.

Reviewer 1 ·

Basic reporting

yes

Experimental design

yes

Validity of the findings

yes

Additional comments

none

Reviewer 2 ·

Basic reporting

All comments have been added in detail to the last section.

Experimental design

All comments have been added in detail to the last section.

Validity of the findings

All comments have been added in detail to the last section.

Additional comments

Review Report for PeerJ Computer Science
(Single-image super-resolution reconstruction based on phase-aware visual MLP)

Thanks for the revision. The new version of the paper and the responses to reviewer comments are generally at an appropriate level. It is observed that the study does not need any additional corrections. Due to its contribution to the literature and in its final form, I recommend that this research paper be accepted. I wish the authors success in their future studies. Kind regards.